# Regional variation of avoidable hospitalisations in a universal health care system: a register-based cohort study from Finland 1996−2013

Kristiina Manderbacka,[1] Martti Arffman,[1] Markku Satokangas,[1,2] Ilmo Keskimäki[1,3]

[1]Service System Research, National Institute for Health and Welfare (THL), Helsinki, Finland
[2]Network of Academic Health Centres and Department of General Practice and Primary Health Care, Helsingin Yliopisto, Helsinki, Finland
[3]Department of Social Sciences, Tampereen Yliopisto, Tampere, Finland

**Correspondence to**
Dr Kristiina Manderbacka; kristiina.manderbacka@thl.fi

## ABSTRACT

**Objectives** A persistent finding in research concerning healthcare and hospital use in Western countries has been regional variation in the medical practices. The aim of the current study was to examine trends in the regional variation of avoidable hospitalisations, that is, hospitalisations due to conditions treatable in ambulatory care in Finland in 1996–2013 and the influence of different healthcare levels on them.

**Setting** Use of hospital inpatient care in 1996–2013 among the total population in Finland.

**Participants** Altogether 1 931 012 hospital inpatient care episodes among all persons residing in Finland identified from administrative registers in Finland in 1996−2013 and alive in 1 January 1996.

**Outcome measures** We examined hospitalisations due to avoidable causes including vaccine-preventable hospitalisations, hospitalisations due to complications of chronic conditions and acute conditions treatable in ambulatory care. We calculated annual age-adjusted rates per 10 000 person-years. Multilevel models were used for studying time trends in regional variation.

**Results** There was a steep decline in avoidable hospitalisation rates during the study period. The decline occurred almost exclusively in hospitalisations due to chronic conditions, which diminished by about 60%. The overall correlation between hospital district intercepts and slopes in time was −0.46 (p<0.05) among men and −0.20 (ns) among women. Statistically highly significant diminishing variation was found in hospitalisations due to chronic conditions among both men (−0.90) and women (−0.91). The variation was mainly distributed to the hospital district level.

**Conclusions** The results suggest that chronic conditions are managed better in primary care in the whole country than before. Further research is needed on whether this is the case or whether this has more to do with supply of hospital care.

## BACKGROUND OF THE STUDY

A persistent finding in research concerning healthcare and hospital use in Western countries has been regional variation in the medical practices. While there are regional variations within countries in disease prevalence, earlier

### Strengths and limitations of this study

► Our register-based data cover the total population in Finland identified from register sources in Finland over a 18-year follow-up period.
► We used two indicators of region of residence that are directly connected to the organisation of public healthcare for the residents enabling us to disentangle the association of each organisational level to regional variations in hospitalisation rates.
► Multilevel models enabled us to capture the hierarchical nature of the data (health centres nested in hospital districts) allowing us to analyse the association more efficiently.
► We were not able to control for need for care, that is, disease prevalence or severity, and thus cannot estimate what part of regional differences in avoidable hospitalisations would be warranted due to differences in disease prevalence and/or severity.

studies have suggested that at least some of the variation in medical care is unwarranted, that is, within country variation does not disappear when need is accounted for.[1] There is also a need to examine whether medical practice variation changes in time.[1] In a relatively recent systematic review, Corallo *et al*[2] report a large number of studies examining medical practice variation from OECD countries covering regional differences in ambulatory care, medical admissions, elective surgery, as well as emergency hospitalisations and avoidable hospitalisations, that is, hospitalisations due to conditions treatable in ambulatory care. Avoidable hospitalisations are commonly used as an indirect indicator of access to and quality of healthcare.[3 4] Earlier studies use slightly different lists of conditions, but generally three types of conditions are examined: admissions that should be avoided with timely access to preventive measures, ambulatory treatment of acute conditions and managing of certain chronic disease in an ambulatory setting.

While there are earlier studies examining regional differences in avoidable hospitalisations, many examine only some parts of the country in question,[5–10] specific age groups,[5 7 10] are cross-sectional[5 9 11 12] and/or study these hospitalisations in specific patient groups, for example, persons with diabetes.[13] Further, some countries publish regularly Atlases of regional differences in medical practices, some of which also report avoidable hospitalisations; see refs [14 15]

The primary aim of this register-based study was to examine trends in avoidable hospitalisations in 1996–2013, whether regional variation increased or decreased in relation to the national trend, that is, possible increase or decrease in regional variation and, secondarily, if so, on what level of healthcare the variation occurred. Few earlier studies have reported development of geographical variation in avoidable hospitalisations with a longitudinal setting over an extensive time period. The Finnish healthcare system provides a good case for examining avoidable hospitalisations as the system is universal in coverage and therefore, in general, supports equal access to both primary and specialised healthcare according to need.[16] The system is mainly financed by tax revenues and user-fees are generally low. The system also supports evidence-based care as there are accepted guidelines for the treatment of altogether 106 conditions in the National Current Care guidelines system.[17] There are ca 160 health centres providing ambulatory primary care in Finland, owned and managed by municipalities themselves or in cooperation with other municipalities. They are the primary producers of ambulatory services for the residents. However, there have been difficulties in access in some areas. There are 21 hospital districts in Finland providing specialist services to the residents. They are owned, funded and managed by federations of municipalities.

## MATERIALS AND METHODS
### The data
We obtained avoidable hospitalisations for the total Finnish population aged 20 years or older in 1996–2013 from the Finnish hospital discharge register, maintained by the National Institute for Health and Welfare (THL). The linkages were performed by competent authorities and the research group received pseudonymised data.

We used the UK definition of avoidable hospitalisations[3 4] with an inclusion of ICD-10 (10th version of International Classification of Diseases) codes J18.9 and J09 (pneumonia). Avoidable hospitalisations were classified into three subcategories: acute, chronic and vaccine-preventable (for diagnoses included and their ICD-10-codes, see online supplementary appendix 1). Hospitalisations were combined if they occurred within 1 day from each other. Data on hospitalisations were linked into demographic data obtained from Statistics Finland.

We used two indicators of region of residence: (1) health centre area (single municipality or cooperation area of municipalities) that is responsible for organising primary healthcare for the residents and (2) hospital district (owned by the federations of municipalities), which is responsible for organising the public specialist services in the area. Publicly funded hospitals produce ca 95% of all specialised inpatient services in Finland. The municipality of residence was linked for each individual at the end of the year preceding hospitalisation. The small autonomous region of Åland was excluded from the analyses. Age at hospitalisation was classified into 5-year age brackets with those 85 years or older as the oldest group. Those living permanently in institutions were excluded from analyses. Total Finnish population was used as the population at risk in the analyses. All analyses were conducted separately for men and women.

### Statistical methods
We calculated national age-standardised rates for each category of avoidable hospitalisations per 10 000 person-years (PYs) with direct standardisation method with the European standard population as standard population. We calculated autocorrelations in each of the categories both in the health centre and in the hospital district level to study consistency in hospitalisation rates and their suitability for further analysis. We found autocorrelations in health centre level vary a lot suggesting that this level was not suitable for the analysis as the result suggests inconsistency in the development in time. The hospital district level autocorrelations were relatively low among men in vaccine-preventable hospitalisations, averaging 0.22, among women the autocorrelations were higher, averaging 0.35. In hospitalisations due to acute conditions, the results were rather similar with autocorrelations averaging 0.44 among men and 0.36 among women. The autocorrelations for hospitalisations due to chronic conditions were higher, with average of 0.78 among both men and women. These numbers suggest that hospital district patterns in hospitalisations were relatively consistent in time and allowed us to examine possible convergence of hospitalisation rates, that is, decrease in the variation, at the hospital district level.

Next, we performed two-level models to receive annual z-scores of the age-standardised hospitalisation rates in each hospital district and category adjusted for polynomial function of time with normally distributed random intercept and slope terms at the hospital district level. Z-scores were then used to examine the hospitalisation rates from different categories of avoidable hospitalisations in a comparable scale. Multilevel models were applied to manage the hierarchical nature of the data and polynomial function of time to capture the general national trend in each category. We calculated correlation between the random intercept and slope terms in these models to obtain an estimate of convergence in hospitalisation rates. A statistically significant negative correlation between random intercepts and random slopes was interpreted as a decrease of regional variation in time. In that case, high random intercepts were correlated with

large negative random slopes, and vice versa, indicating hospital districts approaching the average.

A similar procedure was fitted to all hospitalisation rates due to avoidable causes simultaneously where polynomial function of time was adjusted for each category of avoidable hospitalisations to calculate an estimate for possible general convergence of hospitalisation rates between regions, that is, whether the regional variation in rates decreased more or less in relation to the decreasing national average.[1 18]

In a separate analysis, we used three-level Poisson regression models to examine whether variation between avoidable hospitalisations occurred mainly in the hospital district or health centre level. In these models, year was controlled as a continuous variable and age group as a categorical variable with hospital districts and health centres as random variables.

In order to assess whether there were changes in the levels of healthcare where the regional variation occurred, we estimated models separately for the years 1996–1999 and 2010–2013. Wald test was used to assess the statistical significance of random components in the models.

Also, we calculated intraclass correlation coefficients (ICCs) for the two levels.[19] Variation in age-adjusted avoidable hospitalisation rates in health centre areas was illustrated with scatter plots in the last 4 years of study period. The extremal quotient (EQ) was defined as the ratio of the largest and smallest rates.

### Patient and public involvement

As the study was based on pseudonymised data from administrative registers, patients were not involved in the study. The results will be disseminated to the public through summary articles after the original studies have been published.

### RESULTS

There were altogether 983 966 avoidable hospital admissions among men and 947 046 among women during the study period. More than half of the hospitalisations were due to chronic conditions.

**Table 1** Multilevel model-based correlation between hospital district intercepts and slopes in time between 1996 and 2013

|  | Men | Women |
|---|---|---|
| Preventable | –0.02 ns | 0.44 ns |
| Chronic conditions | –0.90** | –0.91** |
| Acute conditions | –0.45* | 0.01 ns |

ns statistically non-significant.
*P<0.05, **P<0.001.

Figure 1 presents the national rates of avoidable hospitalisations. Avoidable hospitalisations decreased consistently during the study period. The total rate of avoidable hospitalisations was 535 per 10 000 PY among men in 1996 and 306 per 10 000 PY in 2013. Among women, the rates were 340 in 1996 and 221 in 2013. The rates were larger among men throughout the study period in each of the avoidable hospitalisation categories. While the rates remained relatively stable in hospitalisations due to vaccine-preventable conditions and acute conditions, there was a large decrease in hospitalisations for chronic conditions both among men and among women during the study period.

In multilevel analysis (table 1), the correlation between hospital district intercepts and slopes in time was negative among men suggesting decreasing variation when all categories of avoidable hospitalisations were examined together. Among women, no such correlation was found. The correlations were –0.0.46 (p<0.05) among men and –0.22 (ns) among women. However, when examining categories of avoidable hospitalisation separately, negative correlations indicating diminishing variation of hospitalisation rates in hospital districts were found in rates due to chronic conditions among both genders and in rates due to acute conditions among men. In other categories, no correlations indicating change in regional variation were found. The results were mostly similar when health centre level was used in the analysis (data not shown).

Finally, we examined to which regional level (health centre level vs hospital district level) the variation was mainly distributed in each of the categories of avoidable hospitalisations in the beginning (1996–1999) and in the end (2010–2013) of the study period. We examined these two periods due to decrease in avoidable hospitalisation rates during the study period (table 2).

In hospitalisations due to vaccine-preventable conditions, the variance was equally distributed to both levels in the late 1990s and remained similar at the end of the study period among both men and women. In hospitalisations due to chronic conditions, the variance was distributed mainly to the hospital district level. Similar, but smaller variation was found in hospitalisations due to acute conditions. Health centre level variance was also statistically significant in all categories of avoidable hospitalisations among both men and women. However, proportions of

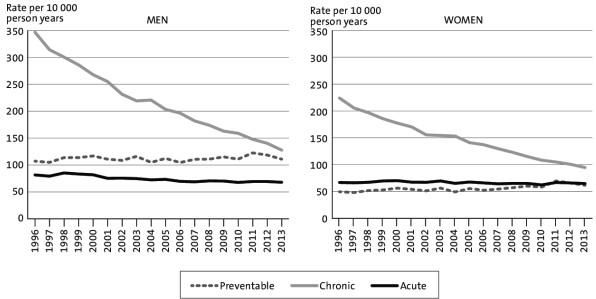

**Figure 1** Trends in avoidable hospitalisations among men and women by category in Finland 1996–2013, age-standardised rates per 10 000 person-years.

**Table 2** The distribution of variance in avoidable hospitalisations to health centre (HC) and hospital district (HD) level in Finland in 1996–1999 and 2010–2013 (Poisson multilevel models controlling for year as a continuous variable and age as a categorical variable; estimates of variance components and their intraclass correlation coefficient (ICC))

| | Category | Level | 1996–1999 | | 2010–2013 | |
| --- | --- | --- | --- | --- | --- | --- |
| | | | Estimate | ICC | Estimate | ICC |
| Men | Preventable | HD | 0.024* | 0.022 | 0.027* | 0.024 |
| | | HC | 0.022** | 0.021 | 0.024** | 0.023 |
| | Chronic | HD | 0.051* | 0.046 | 0.064* | 0.056 |
| | | HC | 0.019** | 0.012 | 0.027** | 0.025 |
| | Acute | HD | 0.025* | 0.022 | 0.039* | 0.034 |
| | | HC | 0.015** | 0.014 | 0.024** | 0.023 |
| Women | Preventable | HD | 0.032* | 0.028 | 0.033* | 0.029 |
| | | HC | 0.033** | 0.031 | 0.029** | 0.027 |
| | Chronic | HD | 0.054* | 0.047 | 0.059* | 0.051 |
| | | HC | 0.031** | 0.029 | 0.033** | 0.030 |
| | Acute | HD | 0.026* | 0.024 | 0.043* | 0.038 |
| | | HC | 0.019** | 0.018 | 0.029** | 0.027 |

*P<0.01, **P<0.001.

variance explained by the two levels remained relatively low according to ICCs.

Figure 2 illustrates health centre-level variation in age-adjusted rates in the three categories of avoidable hospitalisations in the end of the study period (2010–2013). In the figure, each dot represents one health centre. The variation was, in general, smaller among women compared with men in each of the categories. In vaccine-preventable hospitalisations, the health centre rates were close to each other with some outliers especially towards the higher end. EQ among men was 4.80 and among women 4.58. In hospitalisations due to acute conditions, the variation was similar but smaller with an EQ of 4.31 among men and 4.50 among women. In hospitalisations due to chronic conditions, the variation was large, with and EQ of 4.81 for men and 5.36 for women.

## DISCUSSION
### Overview of results
We found a decreasing trend in avoidable hospitalisations from the mid-1990s to the early 2010s. This trend was

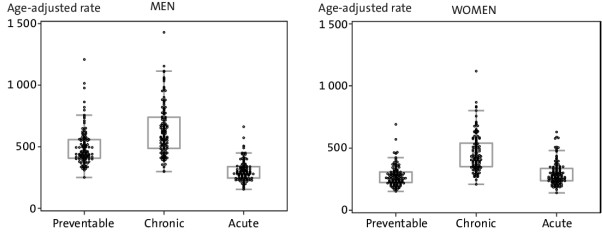

**Figure 2** Health centre-level variation in avoidable hospitalisations among men and women in 2010–2013, age-adjusted rates per 10 000 person-years.

clear in avoidable hospitalisations due to chronic conditions, but not in vaccine-preventable hospitalisations or hospitalisations due to acute conditions. There were large hospital district variations in avoidable hospitalisations due to chronic conditions. Also, in multilevel analyses, the correlation between hospital district intercepts and slopes was negative in these hospitalisations, indicating decreasing hospital district variation and convergence of rates between hospital districts during the study period. Our result concerning relatively large regional variation in avoidable hospitalisations is in line with earlier results from other countries[5–9 11 20 21] and in different categories of avoidable hospitalisations.[14 15] Several reasons have been presented to regional variation, including differences in disease prevalence, rurality, age structure and socioeconomic composition between regions,[6–9 22 23] as well as access to and accessibility of primary healthcare and supply of hospital beds.[9 10 20 21]

Earlier studies have mainly been cross-sectional and have not examined trends in avoidable hospitalisations, but our result concerning decreasing variation is in line with results from Canada.[11] Our study did not include data on disease prevalence or change in it. However, an earlier Finnish study examining persons with diabetes found a similar decrease in hospitalisations due to complications and convergence of hospitalisation rates in the early 2000s suggesting better control of diabetes in primary care.[13] It is possible that disease management improved in other conditions as well thus decreasing the need for specialist hospital care. Further, current care guidelines together with disease-specific programmes may have resulted in more uniform disease management in primary care thus decreasing regional differences in hospitalisations.

Further research is needed to study the effects of differences in the regional sociodemographic composition of the population and supply-side factors to avoidable hospitalisations. On the supply side, an important question is whether the variation is driven by access to and accessibility of primary care or supply of hospital inpatient care. Our results on larger variation in hospital district level in hospitalisations due to chronic conditions support previous results,[6 24] which suggest that different categories of avoidable hospitalisations should be analysed separately.

## Methodological considerations

A major strength of our study was that we were able to use individual-level data of the total population of Finland over a period of 18 years. The data concerning avoidable hospitalisations were derived from the Finnish h ospital discharge register, which has been found to have, in general, good quality and coverage.[25] Use of multilevel models enabled us to capture the hierarchical nature of the data (health centres nested within hospital districts) and the statistical approach allowed us to analyse the development of regional variation in an efficient way. However, even this modern statistical approach is limited in modelling complex phenomenon with considerable individual variation as represented by small ICCs.

Though earlier studies have shown that avoidable hospitalisations associate to factors like need for care, that is, disease prevalence or severity, we were unable to control for them. Thus, we cannot directly evaluate what part of regional differences in avoidable hospitalisations would be warranted due to differences in disease prevalence and/or severity. It is still unclear whether the decrease in rates of the chronic disease category and their regional variation occurred mainly due to the development of primary care or whether it is due to overall reduction in hospital beds. This requires further studies assessing both the development of supply of hospital beds and number of total hospitalisations.

Nor could we evaluate whether primary care has an effect on these differences, which has been strongly debated. Our results of systematically developing regional variation—and its over time division between primary care providers and hospital districts—lay a foundation for further analysis. We consider that analysing how the forementioned factors simultaneously affect the variation of several regional levels and overtime annual trends, poses methodological challenges. This necessitates an independent study with an approach that is optimised to test for these associations. Future studies need also to take into account variation in sociodemographic composition of regions. Our study controlled for gender and age, but not for different dimensions of socioeconomic position.

## CONCLUSIONS

Large but decreasing regional variation was found in our study in avoidable hospitalisations due to chronic conditions. These hospitalisations occur mainly due to complications of the underlying disease. The result may suggest that chronic conditions are better managed in primary care in the whole country than before. Further research is needed on whether this is the case or whether this has more to do with supply of hospital care.

**Contributors** KM contributed to the conception and design of the study, planning of analyses and drafted the manuscript. MA contributed to the conception and design of the study, performed the statistical analyses and took part in the revision of the manuscript for important intellectual content. MS contributed to the conception and design of the study, planning of analyses and took part in the revision of the manuscript for important intellectual content. IK contributed to the conception and design of the study, planning of analyses and took part in the revision of the manuscript for important intellectual content. All authors have read and approved the final manuscript.

**Funding** The study was funded by the Strategic Research Council (grants 312 703 and 312 708) and the Medical Research Council (grant 277 939) at the Academy of Finland.

**Disclaimer** The Academy was not involved in study design, data collection, findings or decision to publish.

**Competing interests** None declared.

**Patient consent for publication** Not required.

**Ethics approval** Ethical approval for the study was received from the Research Ethics Committee of THL (decision #7/2013 §586).

**Provenance and peer review** Not commissioned; externally peer reviewed.

**Data sharing statement** No data are available

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
