## [Reviewer comments · BMJ Open]

ARTICLE DETAILS

TITLE (PROVISIONAL)	Regional variation of avoidable hospitalisations in a universal health care system – a register based cohort study from Finland 1996-2013
AUTHORS	Manderbacka, Kristiina; Arffman, Martti; Satokangas, Markku; Keskimaki, Ilmo

VERSION 1 - REVIEW

REVIEWER	Ester Angulo-Pueyo Aragon Institute For Health Sciences (IACS), Spain
REVIEW RETURNED	04-Mar-2019

GENERAL COMMENTS	I would like to thank for the opportunity of reviewing this manuscript. The analysis of avoidable hospitalisations may help to understand the utilization of healthcare systems and the fact of analysing separately the three categories, results in more meaningful results, as the three groups could stem from different causes. Nevertheless, I think the paper would benefit from further analysis and discussion. A major issue is that authors only superficially discuss and don't investigate into the underlying factors that could explain the observed variation. In that sense, the inclusion of supply and demand side factors in the models could shed light into their interpretation. For example, on the demand side regarding to population needs, burden of disease and socioeconomic level have been associated to these admissions. If authors have access to discharge registers, a proxy indicator composed of admissions reflecting illness differences (rather than coding intensity or supply factors) could be used as a surrogate measure of population burden of disease at hospital district. For example, an indicator formed by hospitalizations for hip fracture, colectomy in colorectal cancer, lung or breast cancer surgery, acute myocardial infarction, and acute ischaemic stroke as it was described in Fisher 2000. In the same line of adjusting for population needs, some morbidity variables (e.g. Elishauser index) could be included in the three level models. There is also evidence about supply factors affecting preventable hospitalisations: hospital offer access to primary care (geographical barriers/distance), volume of primary care professionals or care continuity across healthcare levels, among others. Authors should consider and/or discuss about these variables. For example, if authors have access to discharge registers, variables representing intensity in hospitalisation can be used (Thygesen 2015; Angulo-Pueyo 2017). In general, further and deeper discussion of the results, causes and implications of avoidable hospitalisations variation will enhance the
---

relevance of the study.

Another issue would be related to the second objective which tackles the influence of health care levels in the variation observed. Authors presented the variance due to each level without specifying the proportion of total variance that represents those values and it could occur that those variances represent a negligible amount despite being significant. In order to know the actual relevance of each context in avoidable hospitalisations, the share of the total variation that represents each variance should be assessed. There are some works explaining how to address it in multilevel Poisson models (Austin, 2018).

On the other hand, the Material and Methods section would need a more detailed description of the methodology.

- I would recommend using new paragraphs for each methodology to ease the reading.

- Autocorrelations analysis needs further explanation about its calculation and interpretation. If it is only an analysis to determine if areas are suitable for the analysis, the description in results (page 8 lines 23 to 48) should be moved on to the Methods section. If the analysis implies a result (for example being above or below the national trend) then, the interpretation should be clearly stated in results.

- Analysis of trends in variation also needs further explanation about the model used and the meaning and interpretation of the correlation between the random intercept and slope terms. The sentence "A statistically significant negative correlation was interpreted as a decrease of regional variation in time with higher than average random intercepts linked with a larger than average decrease in random slopes and vice versa" is hard to understand, a rephrasing would be advisable.

- The same applies to the meaning and interpretation of convergence or divergence in regional variation. I assume that may be it refers to increase or decrease in regional variation, but should be explained in methods.

- The sentence "In order to assess possible changes in the levels of regional variation" doesn't exactly describe the objective of assessing the effect of each level in the variation of avoidable hospitalisations.

Other things:

-The objective section in the abstract should include the objective of assessing the influence of each level in the variation. Besides, the phrase "possible convergence or divergence in regional variation" is not clear enough. I would suggest something like "examine trends and influence of different healthcare levels in regional variation in avoidable hospitalisations".

-The background lacks of the specification of the number of health centre areas in Finland and their function, as it has been done with hospital districts.

- In results, the reference to table 1 should be moved after the results of that table, after the sentence "However, when examining categories of avoidable hospitalisations separately.... "

- In results, besides the visual description of figure 2, it could be interesting describe the results with measures of variation such as median, maximum and minimum rate, interquartile range, extremal quotient 5-95, etc.

	References. Fisher et al. Associations among hospital capacity, utilization, and mortality of US Medicare beneficiaries, controlling for sociodemographic factors. Health ServRes2000; 34:1351–62. Thygesen et al. Potentially avoidable hospitalizations in five European countries in 2009 and time trends from 2002 to 2009 based on administrative data. Eur J Public Health. 2015 Feb;25 Suppl 1:35-43 Angulo-Pueyo et al. Factors associated with hospitalisations in chronic conditions deemed avoidable: ecological study in the Spanish healthcare system. BMJ Open. 2017 Feb 24;7(2). Austin, PC et al. Measures of clustering and heterogeneity in multilevel Poisson regression analyses of rates/count data. Stat Med. 2018; 37(4): 572–589).
--	---

REVIEWER	Anne Laurent University of Montpellier
REVIEW RETURNED	06-Mar-2019

GENERAL COMMENTS	The paper addresses an interesting and important topic. It is very well written and structured and easy to read. The authors state the research question and study very clearly. However, it is not clear to which extent the paper contains a scientific contribution. Although the results may be interesting, they could be extended. For instance, the authors could further discuss the limitations of their study. Could they apply predictions, "What if" analysis so as to help the decision makers or to better anticipate? No methodological contribution is provided while there may be some. Have external data been considered to enrich the study? The authors point out some future directions of research but no information is provided, this part could be extended as well.
---

VERSION 1 – AUTHOR RESPONSE

Reviewer: 1

I would like to thank for the opportunity of reviewing this manuscript. The analysis of avoidable hospitalisations may help to understand the utilization of healthcare systems and the fact of analysing separately the three categories, results in more meaningful results, as the three groups could stem from different causes.

1. I think the paper would benefit from further analysis and discussion. A major issue is that authors only superficially discuss and don't investigate into the underlying factors that could explain the observed variation. In that sense, the inclusion of supply and demand side factors in the models could shed light into their interpretation.

For example, on the demand side regarding to population needs, burden of disease and socioeconomic level have been associated to these admissions. If authors have access to discharge

registers, a proxy indicator composed of admissions reflecting illness differences (rather than coding intensity or supply factors) could be used as a surrogate measure of population burden of disease at hospital district. For example, an indicator formed by hospitalizations for hip fracture, colectomy in colorectal cancer, lung or breast cancer surgery, acute myocardial infarction, and acute ischaemic stroke as it was described in Fisher 2000. In the same line of adjusting for population needs, some morbidity variables (e.g. Elishauser index) could be included in the three level models.

There is also evidence about supply factors affecting preventable hospitalisations: hospital offer access to primary care (geographical barriers/distance), volume of primary care professionals or care continuity across healthcare levels, among others. Authors should consider and/or discuss about these variables. For example, if authors have access to discharge registers, variables representing intensity in hospitalisation can be used (Thygesen 2015; Angulo-Pueyo 2017). In general, further and deeper discussion of the results, causes and implications of avoidable hospitalisations variation will enhance the relevance of the study.

OUR RESPONSE 1. We fully understand the need for analyses concerning potential explanations of the regional variation and its' changes over time. However, as there have been few earlier studies examining time trends in regional variation of avoidable hospitalisations, or different levels of health care where the variation occurs, we feel that the trend in the description of variation is an interesting and important issue as such. We feel that analysing how the composition of the population, individual socioeconomic factors, or supply side factors simultaneously affect the variation of several regional levels and over time, poses methodological challenges. This necessitates an independent study with a different methodological approach. We have now added discussion of the matters to the discussion section of the study as suggested by the Reviewer.

2. Another issue would be related to the second objective which tackles the influence of health care levels in the variation observed. Authors presented the variance due to each level without specifying the proportion of total variance that represents those values and it could occur that those variances represent a negligible amount despite being significant. In order to know the actual relevance of each context in avoidable hospitalisations, the share of the total variation that represents each variance should be assessed. There some works explaining how to address it in multilevel Poisson models (Austin, 2018).

OUR RESPONSE 2: The reviewer makes a good point. We have now added a measure of intraclass correlation (ICC) in Table 2 which was calculated according to (Nagakawa et al, 2017). We agree that variances remain relatively low and this is now addressed in the manuscript.

3. On the other hand, the Material and Methods section would need a more detailed description of the methodology.

- I would recommend using new paragraph for each methodology to ease the reading.

- Autocorrelations analysis needs further explanation about its calculation and interpretation. If it is only an analysis to determine if areas are suitable for the analysis, the description in results (page 8 lines 23 to 48) should be moved on to the Methods section. If the analysis implies a result (for

example being above or below the national trend) then, the interpretation should be clearly stated in results.

OUR RESPONSE 3: We have now modified the statistical methods section and added paragraphs. We also moved the autocorrelations analysis to the methods section.

4. Analysis of trends in variation also needs further explanation about the model used and the meaning and interpretation of the correlation between the random intercept and slope terms. The sentence “A statistically significant negative correlation was interpreted as a decrease of regional variation (i.e. convergence) in time with higher than average random intercepts linked with a larger than average decrease in random slopes and vice versa” is hard to understand, a rephrasing would be advisable.

OUR RESPONSE 4: We have now rewritten this part to make it more understandable.

5. The same applies to the meaning and interpretation of convergence or divergence in regional variation. I assume that may be it refers to increase or decrease in regional variation, but should be explained in methods.

OUR RESPONSE 5: We have now rewritten the manuscript to clarify the concepts of convergence and divergence.

6. The sentence “In order to assess possible changes in the levels of regional variation” doesn’t exactly describe the objective of assessing the effect of each level in the variation of avoidable hospitalisations.

OUR RESPONSE 6: We have now amended the sentence pointed out by the Reviewer.

7. The objective section in the abstract should include the objective of assessing the influence of each level in the variation. Besides, the phrase “possible convergence or divergence in regional variation” is not clear enough. I would suggest something like “examine trends and influence of different healthcare levels in regional variation in avoidable hospitalisations”.

OUR RESPONSE 7: We have now amended the objective in the abstract.

8. The background lacks of the specification of the number of health centre areas in Finland and their function, as it has been done with hospital districts.

OUR RESPONSE 8: We have now added text concerning health centre areas in Finland in the background section of the manuscript as indicated by the Reviewer.

9. In results, the reference to table 1 should be moved after the results of that table, after the sentence "However, when examining categories of avoidable hospitalisations separately.... "

OUR RESPONSE 9: Amended as indicated by the Reviewer.

10. In results, besides the visual description of figure 2, it could be interesting describe the results with measures of variation such as median, maximum and minimum rate, interquartile range, extremal quotient 5-95, etc.

OUR RESPONSE 10: We have now added extremal quotient figures in the results section.

References.

Fisher et al. Associations among hospital capacity, utilization, and mortality of US Medicare beneficiaries, controlling for sociodemographic factors. *Health ServRes*2000; 34:1351–62.

Thygesen et al. Potentially avoidable hospitalizations in five European countries in 2009 and time trends from 2002 to 2009 based on administrative data. *Eur J Public Health*. 2015 Feb;25 Suppl 1:35-43

Angulo-Pueyo et al. Factors associated with hospitalisations in chronic conditions deemed avoidable: ecological study in the Spanish healthcare system. *BMJ Open*. 2017 Feb 24;7(2)).

Austin, PC et al. Measures of clustering and heterogeneity in multilevel Poisson regression analyses of rates/count data. *Stat Med*. 2018; 37(4): 572–589).

Reviewer: 2

The paper addresses an interesting and important topic. It is very well written and structured and easy to read. The authors state the research question and study very clearly.

11. However, it is not clear to which extent the paper contains a scientific contribution. Although the results may be interesting, they could be extended. For instance, the authors could further discuss the limitations of their study. Could they apply predictions, "What if" analysis so as to help the decision makers or to better anticipate?

OUR RESPONSE 11: See our response 1.

12. No methodological contribution is provided while there may be some.

OUR RESPONSE 12: We have now modified the methodological considerations section to address the methodological contribution.

13. Have external data been considered to enrich the study?

OUR RESPONSE 13: See our response 1.

14. The authors point out some future directions of research but no information is provided, this part could be extended as well.

OUR RESPONSE 14: We have now extended the part of the discussion concerning future research questions as suggested by the Reviewer.

VERSION 2 – REVIEW

REVIEWER	Ester Angulo-Pueyo Aragon Health Sciences Institute (IACS)
REVIEW RETURNED	30-May-2019

GENERAL COMMENTS	The analysis of avoidable hospitalisations is important for their impact on patients' healthcare quality as well as on health system efficiency. In that sense, this work analysing their evolution (with an appropriate methodology) is an interesting study. However, the mere description of trends in avoidable hospitalisations without any analysis of the factors underlying them, that could help to explain and tackle their variation, limits the relevance of the study to a local context and diminishes its scientific interest.
---